# Subtype of Neuroblastoma Cells with High KIT Expression Are Dependent on KIT and Its Knockdown Induces Compensatory Activation of Pro-Survival Signaling

**DOI:** 10.3390/ijms23147724

**Published:** 2022-07-13

**Authors:** Timofey Lebedev, Anton Buzdin, Elmira Khabusheva, Pavel Spirin, Maria Suntsova, Maxim Sorokin, Vladimir Popenko, Petr Rubtsov, Vladimir Prassolov

**Affiliations:** 1Department of Cancer Cell Biology, Engelhardt Institute of Molecular Biology, Russian Academy of Sciences, 119991 Moscow, Russia; lebedevtd@gmail.com (T.L.); vr.elmira@gmail.com (E.K.); spirin.pvl@gmail.com (P.S.); popenko@eimb.ru (V.P.); rubtsov@eimb.ru (P.R.); 2Center for Precision Genome Editing and Genetic Technologies for Biomedicine, Engelhardt Institute of Molecular Biology, Russian Academy of Sciences, 119991 Moscow, Russia; 3Institute for Personalized Oncology, Sechenov First Moscow State Medical University, 119991 Moscow, Russia; buzdin@oncobox.com (A.B.); suntsova86@mail.ru (M.S.); sorokin@oncobox.com (M.S.); 4Group for Genomic Regulation of Cell Signaling Systems, Shemyakin-Ovchinnikov Institute of Bioorganic Chemistry, 117997 Moscow, Russia; 5Moscow Institute of Physics and Technology, National Research University, 141701 Dolgoprudny, Moscow Region, Russia; 6Department of Bioinformatics and Molecular Networks, OmicsWay Corp., Walnut, CA 91789, USA

**Keywords:** neuroblastoma, ERK kinase, shRNA, lentiviral vector

## Abstract

Neuroblastoma (NB) is a pediatric cancer with high clinical and molecular heterogeneity, and patients with high-risk tumors have limited treatment options. Receptor tyrosine kinase KIT has been identified as a potential marker of high-risk NB and a promising target for NB treatment. We investigated 19,145 tumor RNA expression and molecular pathway activation profiles for 20 cancer types and detected relatively high levels of *KIT* expression in NB. Increased *KIT* expression was associated with activation of cell survival pathways, downregulated apoptosis induction, and cell cycle checkpoint control pathways. *KIT* knockdown with shRNA encoded by lentiviral vectors in SH-SY5Y cells led to reduced cell proliferation and apoptosis induction up to 50%. Our data suggest that apoptosis induction was caused by mitotic catastrophe, and there was a 2-fold decrease in percentage of G2-M cell cycle phase after *KIT* knockdown. We found that *KIT* knockdown in NB cells leads to strong upregulation of other pro-survival growth factor signaling cascades such as EPO, NGF, IL-6, and IGF-1 pathways. NGF, IGF-1 and EPO were able to increase cell proliferation in KIT-depleted cells in an ERK1/2-dependent manner. Overall, we show that KIT is a promising therapeutic target in NB, although such therapy efficiency could be impeded by growth factor signaling activation.

## 1. Introduction

Neuroblastoma (NB) is the most common pediatric extracranial solid tumor responsible for ~15% of all pediatric cancer-related deaths [1]. The prognosis for NB progression and response to treatment can vary dramatically: low-risk disease can be effectively treated and some tumors can even spontaneously regress, whereas the survival rates for refractory or relapsed disease in stage 4 patients are extremely low (~5%), and the standard treatment options for such patients show greatly reduced efficiencies [2,3,4].

Several studies have reported that receptor tyrosine kinase (RTK) KIT (CD117) is expressed in NB tumor cells along with its ligand SCF (stem cell factor) [5,6,7,8]. It was proposed that the KIT-SCF interplay is involved in progression of NB and could be a potential target for tumor therapy [9,10,11,12,13]. In NB tumors and cell lines, KIT-positive cells constitute a sub-population of stem-like cancer cells, and KIT-positive NB cells generate more aggressive tumors than KIT-negative cells [6]. In relapsed patients, stem-like cancer cells showed increased expression of KIT [14,15,16]. Inhibition of KIT activity by different agents, such as specific RTK inhibitors [9,10,11,12,17], targeted antibodies [18], SCF-conjugated bacterial toxins [19], and interfering RNAs [6], leads to decrease in NB cells proliferation rate and tumor growth in vivo [6,9,10,19]. This makes KIT an attractive target for NB therapy, since several drugs, which can target KIT are used for chronic myeloid leukemia (CML), gastrointestinal stromal tumors (GIST) and kidney cancers [20].

Previously, our analysis of growth factor signaling in NB tumors revealed that KIT is predominantly expressed by aggressive tumors with poor outcomes [21]. We showed that RTK inhibitors induce ERK1/2 activation and ERK inhibition improves RTK inhibitor efficacy. Notably, many RTK inhibitors, such as dasatinib, imatinib, cabozantinib, and axitinib, can target KIT, among their other main targets [21]. Since RTK inhibitors can target multiple kinases, it is unclear whether selective KIT inhibition induces the same responses. To better understand whether KIT can be viewed as a primary target for RTK inhibitors and other drugs, we used lentiviral vectors encoding shRNA against KIT to selectively downregulate KIT expression in two neuroblastoma cell lines expressing KIT at remarkably different levels. The SH-SY5Y cells, which have relatively high KIT expression, were used to evaluate the effects of KIT knockdown on cell survival and the associated changes in molecular signaling. In contrast, the SK-N-AS cells showing relatively low expression of KIT, were used as the non-specific control to identify effects of KIT downregulation on cell survival, and also as the control for any possible off-target effects of anti-KIT small hairpin RNA (shRNA). We examined how changes in growth characteristics (proliferation rate, proportion of apoptotic cells and cell cycle progression) correlated with the activities of intracellular molecular pathways.

## 2. Results

### 2.1. KIT Expression Hallmarks in NB

First, we used several datasets to analyze different aspects of KIT expression in NB: expression patterns relative to other malignant diseases and association with immune infiltration; NB cells dependency on KIT expression, association with clinical features; and correlations with the activity of signaling pathways (Appendix A). We used gene expression data for 19,145 patient samples from 144 datasets covering 20 cancer types (R2: Genomics Analysis and Visualization Platform, http://r2.amc.nl, accessed on 29 March 2022) and applied distribution clustering (Figure 1A, Appendix A). This clustering method combined tumors into groups with similar KIT expression distribution patterns among tumors. For example, acute myeloid leukemia (AML), myeloma, and Ewing sarcoma have a substantial percentage of samples with high KIT expression, while chronic lymphoblastic leukemia (CLL) has low mean KIT expression, and no patients with CLL have high KIT expression. Neuroblastoma, overall, has an average mean KIT expression similar to most other solid tumors, but like kidney, lung, breast, medulloblastoma, ependymoma, and CNS/PNET tumors, it has a substantial percentage (>5%) sample with high KIT expression (higher than mean KIT expression in AML).

Since in bulk tumor samples gene expression can be affected by tumor microenvironment and *KIT* can be expressed by immune cells, we analyzed the correlation between *KIT* expression and typical immune infiltration signatures for different cancer types (Appendix A) [22]. This analysis revealed three cancer types: cancers that have a high number of positive correlations between *KIT* expression and immune signatures, cancers with no significant correlations, and some cancers had a negative correlation (Figure 1B). For tumors like breast cancer, where *KIT* expression strongly correlates with immune infiltration signatures, *KIT* is most probably mainly expressed by the infiltrating immune cells. On the other hand, for tumors with a few or no significant correlations (*p* < 0.05 after FDR correction and |R| > 0.2), including neuroblastoma, CNS/PNET, ependymoma, medulloblastoma, lung cancer, and myeloma, KIT is most likely expressed by the tumor cells themselves (Figure 1A,B). Kidney cancer, unlike all other solid cancers, has many negative significant correlation records for KIT expression and immune infiltration signatures, however the possible reason for such a phenomenon is unclear.

Next, we analyzed the gene dependency data from the Cancer Dependecy Map (DepMap) for KIT depletion by RNAi [23]. NB cells showed the highest sensitivity to KIT depletion by RNAi among solid tumor types (Figure 1C), meaning that KIT is a promising drug target specifically for NB cells. Most approved KIT inhibitors have multiple other targets, so we analyzed the correlation between NB cells sensitivity to FDA-approved kinase inhibitors, which have KIT among their targets, and the dependency of NB cells on these targets according to DepMap RNAi and CRISPR screens [24,25]. FDA-approved drugs (imatinib, dasatinib, sunitinib, midostaurin, pazopanib, sorafenib, and tivozanib) and their targets were selected from CHEMBL database [26] and drug sensitivity data were taken from PRISM drugs screen study [27]. Among known drug targets KIT and RET showed highest correlation with sensitivity to kinase inhibitors (Figure 1D), so NB cells that are sensitive to these inhibitors are also sensitive to KIT depletion. As was previously shown, KIT expression is higher in MYCN-amplified and INSS stage 4 tumors, and in patients older than 18 months [6,8,9,21]. These patterns of KIT expression were confirmed by analysis of the integrated Cangelosi dataset (Appendix A). Additionally, higher KIT expression in neuroblastoma was associated with poor disease outcomes (Figure 1E), as has been shown in previous studies [6,21]. Overall, these data indicate that KIT is potentially one of the main targets for multikinase inhibitor drugs in NB tumor cells and is relevant for more aggressive tumors.

To further characterize possible KIT functions in neuroblastoma we analyzed the correlation of *KIT* expression and activation levels of signaling pathways, calculated using the Oncobox method [28] for 60 previously published NB tumor RNA expression profiles [21,29]. We found 117 signaling pathways with significant correlation between pathway activation strength (PAS) and *KIT* expression level (*p* < 0.05 after FDR correction) (Appendix A). Tumor infiltration and signaling pathway for immune modulating cytokine IL-2 had a strong negative correlation with *KIT* expression, which is in line with our conclusion that *KIT* expression in neuroblastomas is not driven by immune infiltration (Figure 1C). In turn, *KIT* expression showed a positive correlation with the *ERK cell survival* signaling pathway and negative with *ATM cell G2-M cell cycle arrest* pathway, thus suggesting a possible role of KIT in neuroblastoma cell proliferation, survival, and control of G2/M cell cycle checkpoint (Figure 1C). We also found statistically significant negative correlations for several apoptosis-related pathways, including *mitochondrial apoptosis* pathway, although the role of KIT in control of neuroblastoma cells apoptosis needs to be further investigated (Figure 1C, Appendix A). Interestingly, *KIT* expression negatively correlated with most of growth factor signaling pathways, such as erythropoietin (EPO), nerve growth factor (NGF), epidermal growth factor (EGF), hepatocyte growth factor (HGF), and vascular endothelial growth factor (VEGF) pathways (Figure 1C, Appendix A). This potentially indicates that activation of KIT/SCF pathway and certain growth factor pathways are mutually exclusive.

### 2.2. KIT Knockdown Induces Cell Death via Increased Apoptosis

To investigate KIT role in the maintenance of NB cells we downregulated KIT expression using lentiviral vectors expressing shRNA against KIT mRNA (shKIT) (Figure 2A). Previously we characterized KIT expression in NB cell lines [7], and for this experiment we selected KIT-overexpressing NB cells SH-SY5Y and NB cells with weak KIT expression SK-N-AS. We selected SK-N-AS cells as the control, which should be dependent on KIT, and can allow us to discriminate off-target effects caused by lentiviral transduction or shRNA off-targets. We were able to downregulate *KIT* in both cell types (Figure 2B and Appendix A), and KIT knockdown in SH-SY5Y cells was further confirmed by immunocytochemistry analysis (Figure 2C and Appendix A). KIT is expressed at the protein level in NB tumors [8], and we previously showed that KIT mRNA levels had a good association with KIT protein levels in NB cells [7]. These data are supported by a strong correlation between KIT mRNA and protein levels in NB cells according to CCLE data (Figure 2D) [30]. In SH-SY5Y cells, *KIT* knockdown led to significant inhibition of cell proliferation for at least 14 days (Figure 2E). Additionally, *KIT* knockdown in SH-SY5Y cells induced apoptosis 3 days after shRNA lentiviral transduction and apoptosis further increased on day 6 (Figure 2F). Notably, *KIT* downregulation in SK-N-AS cells did not affect their proliferation rate and did not cause apoptosis induction. This indicates that shKIT induces cell death in SH-SY5Y mainly through *KIT* downregulation and not due to lentiviral vector off-target effects, as it did not affect KIT-negative SK-N-AS cells.

To characterize KIT functions in NB cells, we performed transcriptome analysis using custom microarrays and OncoBox algorithm to analyze which molecular pathways are affected by *KIT* knockdown. Overall, we analyzed activities of 271 signaling and 321 metabolic pathways in SH-SY5Y and SK-N-AS cells 3 and 6 days after shKIT lentiviral transduction (Appendix A). Consistent with our previous results, activities of 56% of metabolic pathways were remarkably reduced in SH-SY5Y cells after 6 days of *KIT* knockdown, when the most prominent effects on cell survival and proliferation were detected (Figure 2G). In the control SK-N-AS cells, only ~1% of the pathways were downregulated 6 days after knockdown of *KIT*.

### 2.3. KIT Knockdown in SH-SY5Y Cells Induces Cell Cycle Arrest

On the other hand, signaling pathway analysis showed that *KIT* knockdown in SH-SY5Y cells caused upregulation of several DNA repair and S-phase cell cycle transition pathways (Figure 3A). To verify potential changes in cell cycle transition, we measured expression changes for several major cell cycle regulators after KIT knockdown. Expression of cyclins A1, B1 and E1 (*CCNA1*, *CCNB1* and *CCNE1*) were downregulated at both time points, while on day 6, expression of cyclin E1 decreased even further (Figure 3B). Cyclin A1 and B1 are major regulators of G1/S and G2/M progression and their downregulation is associated with cell cycle arrest, mitotic catastrophe, and delayed apoptosis. Additionally, we observed increased expression of the tumor suppressor gene p27^Kip1^ (*CDKN1B*) (Figure 3B). Downregulation of *MYC* expression, that intensified over time may have a direct impact on the cell cycle progression as MYC is the positive regulator of cyclins A1 and E1, and the negative regulator of p27^Kip1^ expression [31].

We further verified changes in cell cycle transition using SYTOX DNA staining and flow cytometry (Figure 3C). On day 3, we observed an increased percentage of SH-SY5Y cells in G2/M phase (35% for shKIT vs. 28% for shSCR). Since we observed inhibited proliferation and increased apoptotic rate (Figure 2B,C), thus suggesting that the cells that entered G2 phase by day 3 could not progress cell cycle further and entered mitosis. At day 6, the proportion of G2/M SH-SY5Y shKIT cells dropped dramatically (9% for shKIT vs. 24% for shSCR) while the proportion of cells in the S-phase increased (20% for shKIT vs. 11% for shSCR). Overall these changes in cell cycle progression were consistent with the activation of signaling pathways related to S-phase transition (Figure 3A) and downregulation of *CCNA1* and *CCNB1* expression (Figure 3B).

Although activities of cell cycle pathways were also moderately increased in both SK-N-AS and SH-SY5Y cells, by day 6, activities of signaling pathways in SK-N-AS cells decreased, while in SH-SY5Y cells, their activity changes intensified (Figure 3A). Consistent with no effect of shKIT on SK-N-AS proliferation, there were no considerable changes in the expression of cell cycle regulator genes or distribution of cell cycle phases (Appendix A).

### 2.4. ERK Pathway Activation Is Essential for NB Cell Survival after KIT Knockdown

Although KIT is known to activate a number of intracellular kinases, such as ERK1/2, AKT and JAK2, paradoxically KIT knockdown caused an increase in the activity of most intracellular kinase pathways (Figure 4A). We hypothesized that kinase activation may be a compensatory response to KIT knockdown to promote cell survival. ERK/MAPK [21,32,33], JAK/STAT [34,35], and PI3K/AKT [36] are considered promising pathways for targeting aggressive NB cells. We tested how ERK1/2 (FR180204), JAK2 (AG490) and PI3K/AKT (wortmannin) inhibitors affected SH-SY5Y proliferation after KIT knockdown. We selected non-toxic and moderately toxic concentrations for each inhibitor and added them on day 3 after shRNA lentiviral transduction, and then on day 6 we changed the growth medium and added inhibitors in the same concentrations (Figure 4B). ERK1/2 inhibition significantly reduced SH-SY5Y shKIT proliferation, even when non-toxic concentration (for control cells) was used. In turn, JAK2 and PI3K/AKT inhibitors had a smaller impact on SH-SY5Y shKIT proliferation (Figure 4B).

We then used previously established ERK-KTR SH-SY5Y [21,33,37] and generated SK-N-AS reporter cells to measure how these inhibitors and *KIT* knockdown specifically affected ERK1/2 activity. Briefly, ERK-KTR is a kinase translocation reporter [38], which allows the measurement of ERK activity in live cells by calculating the reporter fluorescence ratio in cell’s cytoplasm and nuclei. KIT knockdown increased ERK activity in SH-SY5Y cells and did not affect ERK1/2 in SK-N-AS cells (Figure 4C). As expected ERK1/2 inhibitor FR180204 reduced ERK activity in a dose-dependent manner (Figure 4D). JAK2 inhibitor AG490 had no effect on ERK activity and wortmannin reduced ERK activity only at 10 μM concentration which displayed toxicity towards both SH-SY5Y shSCR and shKIT cells. These results indicate the importance of ERK1/2 activation in response to *KIT* knockdown for NB cell survival.

### 2.5. Compensatory Activation of Growth Factor Signaling Rescues Cells from KIT Knockdown

Similar to activation of major intracellular kinases KIT knockdown caused activation of growth factor signaling pathways, such as EPO, IGF-1, WNT, NGF, IL-6, HGF, and ErbB ligands (Figure 5A). WNT pathway was also strongly upregulated in SK-N-AS cells, suggesting that its activation could be caused by lentiviral transduction or introduction of shRNA itself; however, this activation dramatically decreased by day 6. Expression of growth factor receptor genes *EPOR*, *NTRK1* (encodes NGF receptor TRKA), *IL6R* and *IGF1-R*, was also increased 6 days after transduction (Figure 5B). Expression of *EPOR*, *IL6R* and *IGF1-R* was also increased in SK-N-AS cells, although the magnitude of these changes was considerably lower (Figure 5B).

To check if these growth factors affect cell survival, we treated SH-SY5Y cells after *KIT* knockdown with recombinant growth factors. Recombinant growth factors were added on day 3 after shKIT lentiviral transduction, then on day 6, the growth medium was changed and growth factors were added in the same concentrations, and this process was repeated on days 9 and 12. EPO, NGF, IGF-1, and IL-6 stimulated SH-SY5Y shKIT cells proliferation (Figure 5C); however, IL-6 had a significant effect only after third treatment at day 12, and IGF-1 had strong effect on control cell proliferation (Appendix A). EPO and NGF also reduced apoptosis induction by KIT downregulation (Figure 5D). SH-SY5Y cells after *KIT* downregulation became sensitive to ERK1/2 inhibition and ERK1/2 are the main kinases activated by growth factors. ERK1/2 inhibition in SH-SY5Y shKIT blocked EPO and NGF action on cell survival and apoptosis (Figure 5E). To verify these findings, we selected two additional NB cell lines with relatively high KIT expression: SK-N-BE(2) and IMR-32 (Figure 2D). Both cell lines were sensitive to KIT knockdown and EPO rescued both SK-N-BE and IMR-32 cells (Appendix A). *EPOR*, *NTRK1*, *IL6R*, and *IGF1R* expression was upregulated in SK-N-BE(2) cells (Appendix A). Due to high toxicity of KIT knockdown for IMR-32 cells we were not able to measure gene expression changes in these cells.

## 3. Discussion

In this study, we showed that KIT might play a major role in regulation of apoptosis and cell cycle progression for NB cells. By comparing pathways that show a strong correlation with *KIT* expression in NB tumors and pathways affected by *KIT* knockdown in NB cells, we identified molecular processes most likely associated with KIT functions in vivo in NB tumors. Thus, *KIT* expression negatively correlated with cell cycle G2-M checkpoint arrest and *KIT* knockdown caused upregulation of signaling pathways related to cell cycle S-phase, and we detected a significant decrease in the percentage of G2 phase cells, and accumulation of S phase cells. *KIT* expression also positively correlated with cell survival pathways and negatively with several apoptosis induction pathways, while *KIT* knockdown caused strong induction of apoptosis. This indicates that high *KIT* expression in NB tumors prevents apoptosis induction and stimulates mitosis, thus contributing to more aggressive tumor growth. Moreover, in DepMap dependency screens, NB showed the highest dependency on KIT among solid tumor cells. In addition, our pan-cancer analysis showed that some cancers, for example, kidney, lung cancer, medulloblastoma, and CNS/PNET cancers have hallmarks of *KIT* expression similar to NB: high *KIT* expression, which is not associated with immune infiltration. This suggests a possible pro-oncogenic role of KIT in these tumors, which is partially supported by other published results [39,40,41]. Early histological analyses indicated that higher KIT content in NB tumors might be associated with less aggressive tumors and a more favorable prognosis [42,43,44]. However, later studies showed a strong association of KIT with cancer stem cells, stage 4, and MYCN-amplified tumors [6,8,9,14,16,45,46], and our data strongly support these findings.

Previously, we showed that RTK inhibitors, such as imatinib, cause increased expression of several growth factor receptors, including *EPOR* and TRKA, and elevate ERK1/2 activity in NB cells [21]. Here, unbiased dependency analysis showed that KIT is probably one of the main targets for several RTK inhibitors, and we observed effects after *KIT* knockdown by shRNA similar to those induced by RTK inhibitors [21]: increase in *EPOR*, *NTRK1*, *IL6R*, and *IGF1R* expression and ERK1/2 activity. Moreover, *KIT* expression negatively correlated with activities of several growth factors signaling pathways, including EPO, NGF, EGF, and HGF, while *KIT* knockdown led to upregulation of these signaling pathways, which promoted cell survival and blocked apoptosis induction. It has been shown that EPO and SCF may have a synergistic effect on erythropoiesis [47] and migration of cervical cancer cells [48], and *EPOR* and KIT may even form a receptor complex [49]. However, in erythroleukemic cells, EPO downregulates *KIT* expression [50] and malignant transformation of melanocytes leads to a loss of KIT expression and upregulation of *EPOR* expression in melanoma cells [51]. Notably, melanocyte lineage is derived from neural crest cells, which also give rise to NB cells, and we detected similar negative correlation between *KIT* expression and EPO signaling activity. This indicates that malignant transformation may affect the relation between *EPOR* and KIT, and the exact mechanism needs further investigation.

Here, we also found a strong decrease in *MYC* expression after KIT knockdown (Figure 3B). MYC is a potent regulator for many receptors gene transcription. A study by Duncan J. et al. shows that inhibition of RTK signaling including ERK1/2 leads to MYC degradation, which triggers kinome reprogramming and increases RTKs expression [52]. This then leads to reactivation of ERK1/2 activity and cell adaptation to initial stress. Additionally, MYC/MAX complex may act as a component of a negative feedback control of ERK1/2 activity. MYC is activated by ERK1/2 phosphorylation [53] and MYC/MAX complex activates expression of dual-specificity phosphatases DUSP2 and DUSP7 [54], which negatively regulate ERK1/2 phosphorylation. Additionally, many growth factor receptors have common regulatory proteins, which connect them with MYC and MAPKs as shown by GeneMania analysis (Appendix A). These adaptor proteins might also be involved in RTK reprogramming. However, these speculations on mechanisms of kinome reprogramming in NB cells are not experimentally validated and need further experimental confirmation.

Since NB cell response to dasatinib or imatinib is similar to *KIT* knockdown, we hypothesize that the compensatory response induced by these drugs may also be due to KIT inhibition, which induces a switch to alternative signaling through, e.g., *EPOR* and TRKA. Alternatively, an increase in EPO and NGF signaling might be explained by cell heterogeneity and clonal expansion of cells that predominantly express *EPOR* and TRKA over KIT. Overall, we show that KIT has an important role in the regulation of apoptosis, mitosis, and survival of KIT-positive NB cells and tumors. *KIT* knockdown induces considerable changes in growth factor signaling, and some of these changes potentially allow NB cells to survive *KIT* downregulation. These changes, especially ERK1/2 upregulation, should be considered for tailoring new therapeutic approaches for targeting KIT in neuroblastoma, and potentially in other types of cancer.

## 4. Materials and Methods

Cell cultures. Human neuroblastoma cells without MYCN-amplification SH-SY5Y (ATCC CRL-2266) and SK-N-AS (ATCC CRL-2137), and MYCN-amplified SK-N-BE(2) (ATCC CRL-2271) and IMR-32 (ATCC CCL-127) were cultured in RPMI 1640 medium (Gibco, Billings, MT, USA) supplemented with 10% fetal calf serum (FCS) 100 units/mL penicillin, 100 μg/mL streptomycin and 1 mM sodium pyruvate at 37 °C and 5% CO_2_. HEK293T cells were used for generation of lentiviral particles stocks and were cultured in DMEM medium (Gibco) supplemented with 10% fetal calf serum (FCS) 100 units/mL penicillin, 100 μg/mL streptomycin and 1 mM sodium pyruvate at 37 °C and 5% CO_2_. All cell lines were gifted by the Heinrich-Pette Institute—Leibniz Institute for Experimental Virology. None of the used cell lines is listed in the list of commonly misidentified cell lines maintained by the International Cell Line Authentication Committee and all cell lines were tested for mycoplasma contamination.

Lentiviral pseudotyped particles production. The DNA sequences encoding anti- KIT or non-specific control SCR small hairpin RNAs (shRNAs) were subcloned into the HpaI/XhoI sites of LeGO-C plasmid containing the mCherry protein. LeGO-C plasmid for shRNA expression was a gift from Boris Fehse (http://www.lentigo-vectors.de/, accessed on 29 May 2022) [55]. pLentiCMV Puro DEST ERKKTRClover was a gift from Markus Covert (Addgene plasmid # 59150) [38]. Sequences of both shSCR and shKIT were previously described [56,57]. The stocks containing VSV-G pseudotyped lentiviral particles were generated by transfection of 2 × 10^6^ HEK293T cells plated in 10 cm dishes with 10 μg LeGO-C or pLentiCMV Puro DEST ERKKTRClover, 10 μg gag-pol pMDLg/pRRE plasmid, 5 μg pRSV-Rev plasmid and 2 μg pVSV-G plasmid using calcium phosphate transfection ProFection kit (Promega, Madison, WI, USA). Then, 6 h after transfection, the medium was changed with DMEM containing 20 mM HEPES. Lentivirus-containing supernatants were collected twice every 24 h after transfection. Collected lentiviral stocks were filtered through 0.22 µm nitrocellulose filter and stored at −80 °C.

Titration of lentiviral stocks and transduction. Lentiviral stocks were initially titrated on HEK293T to evaluate titer. HEK293T cells were seeded in 24-well plate, 3 × 10^4^ cells per well. On the next day, different volumes of lentiviral stocks (1–10 μL) were added to the cells. Then, 72 h after transduction, rates were measured by LSRFortessa flow cytometer. Cells positive for fluorescent protein mCherry (LeGO shRNA lentiviral vectors) and for mClover (pLentiCMV Puro DEST ERKKTRClover) were determined as positively transduced. Titer (lentiviral particles per ml of supernatant) was calculated for each lentiviral stock using volumes of stock where transduction was still linear (10–30%). For transduction of neuroblastoma cells, the same amount of each shRNA lentiviral vector was used. To determine working amounts of lentiviral particles, we performed titration on SH-SY5Y and SK-N-AS cells in the same manner as for HEK293T cells. The amount of shRNA lentiviral particles per cell that resulted in at least 90% transduction was used for further experiments (Appendix A). For creation of SH-SY5Y cells expressing ERK KTR, cells were transduced with ERKKTRClover lentiviral particles to achieve ~30% transduction rate. Then, 72 h after transduction, 1 μg/mL puromycin (Sigma, St. Louis, MO, USA) was added. After one week, the selection was confirmed on a fluorescence microscope. Transduction rates were verified for each time point used in the experiments.

Immunocytochemistry. SH-SY5Y cells were plated on a glass 24 h prior lentiviral transduction. The amount of lentiviral particles used for transduction was adjusted accordingly to a number of cells plated. Then, 72 h after transduction with shSCR or shKIT lentiviral particles cells fixed with 4% paraformaldehyde (PFA) in PBS for 10 min, washed with PBS three times and incubated overnight at 4 °C with FITC-conjugated anti-KIT antibodies (ab119107, Abcam, Cambridge, UK) diluted 1:50 in 1% BSA in PBS buffer. FITC-conjugated mouse IgG1 antibodies (ab91356, Abcam, Cambridge, UK) were used as a control. Then, cells were washed with PBS and were mounted in Slowfade gold medium (Invitrogen, Waltham, MA, USA) containing 1 μM DAPI (Sigma-Aldrich, St. Louis, MO, USA), and sealed with nail polish. Digital images were obtained using a Leica TCS SP5 laser-scanning microscope (Leica, Wetzlar, Germany) equipped with an HCX PLAPO CS 63 × 1.4 oil immersion lens. The image acquisition parameters were as follows: DAPI fluorescence (DNA staining) with excitation at 405 nm and emission at 420–480 nm; FITC-conjugated antibodies fluorescence: excitation at 488 nm, emission at 500–560 nm, mCherry fluorescence: excitation at 543 nm, emission at 580–620 nm. Images were processed using the same parameters on LAS AF Lite software (Leica).

Analysis of cell growth, apoptosis and cell cycle. The cells were counted on Neubauer chamber by trypan blue exclusion method. Apoptosis was measured at the same time points by double staining with annexin V-FITC (ThermoFisher, Waltham, MA, USA) and SYTOX Blue (ThermoFisher). Cells positive for annexin V-FITC staining and negative for SYTOX Blue staining were selected as apoptotic cells. All measurements were performed on LSRFortessa flow cytometer (BD Biosciences, San Jose, CA, USA) and analyzed with FlowJo software. Flow cytometry was used to detect the cell cycle distribution. The cells were fixed with ice-cold 70% ethanol at 4 °C overnight and stored at −20 °C no longer than for one week until the measurement was done. On the day of the measurement the cells were washed twice with PBS, treated with 100 μg/mL RNase-A (Sigma) and stained with SYTOX Blue (Thermo Scientific, Waltham, MA, USA). All measurements were performed on LSRFortessa flow cytometer (BD Biosciences) and analyzed with ModFit LT 5.0 software.

Growth factor and inhibitors treatment. Then, 72 h (3 days) after lentiviral transduction of neuroblastoma cells, the medium was changed to medium containing either 100 ng/mL growth factors, kinase inhibitors, or their combination. For experiments where cells were treated with FR180204 and growth factors, cells were pretreated with FR180204 two hours prior addition of growth factors. FR180204, AG490 and wortmannin were diluted in DMSO and appropriate amounts of DMSO were used as control. For prolonged treatment with inhibitors growth medium was changed every 3 days and fresh inhibitors added at the same concentrations. FR180204 and AG490 were purchased from Sigma-Aldrich (St. Louis, MO, USA), wortmannin was purchased from Selleckchem, and recombinant EPO, NGF, HGF, IGF-1, EGF and IL-6 were purchased from Abcam (Cambridge, MA, USA).

Quantitative Real-time PCR. RNA extraction was performed using TRIzol reagent (Invitrogen) in accordance to manufacturer’s protocol on third and sixth days after transduction. Then, 2 μg of RNA were used for the synthesis of cDNAs. Real-time PCR was performed in five replicates using the Maxima SYBR Green Supermix (Thermo Scientific, Waltham, MA, USA) and CFX96 Real-Time System (Bio-Rad, Hercules, CA, USA). The expression levels of studied genes were normalized to that of the human *GAPDH*, Ct values and relative expression was determined by CFX Manager 3.1 software (Bio-Rad, Hercules, CA, USA). Primer sequences are presented in Appendix A.

Synthesis of microarrays. B3 microarray synthesizer (CustomArray, Bothell, WA, USA) was used for forty nucleotides-long oligonucleotide probe synthesis on CustomArray ECD 4 × 2K/12K slides. Synthesis was performed according to the manufacturer’s recommendations. Two replicates of total 6020 unique oligonucleotide probes specific to 3706 human gene transcripts were placed on each chip. Chip design was performed using Layout Designer software (CustomArray, Bothell, WA, USA). For the custom microchip, we used original oligonucleotide probe sequences of the Illumina HT 12 v4 platform.

Library preparation and hybridization. Complete Whole Transcriptome Amplification WTA2 Kit (Sigma) was used for reverse transcription and library amplification. Manufacturers protocol was modified by adding to amplification reaction dNTP mix containing biotinylated dUTP, resulting to final proportion dTTP/biotin-dUTP as 5/1. Microarray hybridization was performed according to the CustomArray ElectraSense™ Hybridization and Detection protocol. Hybridization mix contained 2.5 µg of labeled DNA library, 6X SSPE, 0.05% Tween-20, 20 mM EDTA, 5× Denhardt solution, 100 ng/uL sonicated calf thymus gDNA, 0.05% SDS. Hybridization mix was incubated with chip overnight at 50 °C. Hybridization efficiency was detected electrochemically using CustomArray ElectraSense™ Detection Kit and ElectraSense™ 4 × 2K/12K Reader.

Initial processing of microarray data. Probe signals were geometrically averaged, thus obtaining expression value for each specific type of the probe. Then, quantile normalization was performed using the ‘preprocessCore’ R package (https://rdrr.io/bioc/preprocessCore/, accessed on 30 June 2018) [58]. Gene expression data were deposited in Gene Expression Omnibus database with the accession number GSE116175.

Pathway activation strength calculation. Based on normalized gene expression values we calculated pathway activation levels for molecular pathways from the Oncobox collection of molecular pathways [59]. For each molecular pathway, we calculated pathway activation level using previously described formula [28]. For each molecular pathway, we calculated pathway activation level using the following formula [28]:*PALp* = ∑*NIInp* × *ARRnp* × log*CNRn*∑*NIInp* × |*ARRn*|*n/n*
where *PALp*—molecular pathway *p* activation level; *CNRn* (case-to-normal ratio)—ratio of the protein-encoding gene n product concentrations in the test sample and in the norms (average value in the control group); *NIInp*—index of gene product n assignment to the pathway *p*, assuming the values equal to 1 for gene products included in the pathway and equal to 0 for gene products not included in the pathway; discrete value *ARRnp* (activator/repressor role) is deposited into the molecular pathway base and determined for a gene *n* in the pathway *p* as follows: *ARRnp*= {−1; protein *n* is a signal repressor in a pathway *p* − 0.5; protein *n* is more likely a signal repressor in a pathway *p*0; the role of a protein *n* in a pathway *p* is either ambivalent or neutral 0.5; protein *n* is a signal activator in a pathway *p*1; protein *n* is a signal activator in a pathway *p*}.

Pan-cancer analysis and immune signature score calculation. Data for 19,145 patient samples from 144 datasets was obtained from R2: Genomics Analysis and Visualization Platform, http://r2.amc.nl, accessed on 29 March 2022. Only datasets using human genome coverage u133p2, MAS5.0 normalization and number of samples larger than 50 were included in the analysis. Tumor gene expression data was combined into 20 cancer types (Appendix A). Distribution analysis, Ward.D2 clustering, and heatmaps generation were performed using ComplexHeatmap R package [60]. Pan-cancer immune signatures were obtained from [22] and signatures for each sample were calculated as a mean squared expression for all genes included in this signature. Pearson correlation and two-stage FDR correction were calculated using Statsmodels Pyhton package.

Analysis of the Cancer Dependency Maps data. For comparison of tumor cell types gene scores were taken from RNAi (Achilles+DRIVE+Marcotte, DEMETER2) DepMap dataset [23,24]. Cell lines present in the analysis were grouped by primary disease (tumor) type according to CCLE cell line description. Overall, 24 tumor types represented by more than 5 cell lines were used for comparison. For drug sensitivity-target dependency correlation analysis, we used gene scores from RNAi (Achilles+DRIVE+Marcotte, DEMETER2) and CRISPR (DepMap 22Q2 Public+Score, Chronos) [25] DepMap screens. FDA-approved drugs and their targets were selected from CHEMBL database [26], and drug sensitivity data (area under curve) was taken from PRISM drugs screen study [27]. For each drug–target pair, correlation coefficients were calculated separately for RNAi and CRISPR screens and then averaged. Drug targets arranged by their mean correlation coefficients and drugs are arranged by Ward.D2 clustering using ComplexHeatmap R package [60]. Pearson correlations were calculated using Statsmodels Pyhton package.

Microscope imaging and ERK KTR quantification. SH-SY5Y and SK-N-AS cells expressing ERK KTR were seeded into 96-well plates, on the next day, transduced with shRNA lentiviral vectors and on the third day after transduction images were taken. For measuring kinase inhibitors impact on ERK activity, cells were seeded, on the next day inhibitors were added and images taken 24 h after treatment. For nuclear segmentation cells were incubated with 100 ng/mL Hoechst-33342 for one hour before imaging. Each experiment was repeated six times, two microscopic fields with appropriate densities were chosen for imaging for each well. Images were taken on Leica DMI6 fluorescence microscope with 10x objective. Around 200–300 individual cells for each microscopic field were analyzed and cytoplasm to nucleus ratios (C/N ratio) of mClover intensity were calculated for each cell. Illumination correction, segmentation and object intensity calculations were performed with CellProfiler [61]. Cytoplasm was segmented for each nucleus using Otsu thresholding method. Median intensities of mClover fluorescence in cytoplasm and nucleus were quantified and used to calculate cytoplasm to nucleus (C/N) ratios for each cell. Higher C/N ratio represents higher ERK activity.

Statistical analysis. All the data are expressed as mean ± SD from at least three individual experiments, unless stated otherwise in the text. Statistical significances of differences observed in cell viability experiments were determined by Mann–Whitney non-parametric test. Statistical significances for real-time PCR experiments were determined by unpaired two-sided Student *t*-test. Statistical calculations and data processing were performed in Python 3.7 and GraphPad Prism 9 software.

## Figures and Tables

**Figure 1 ijms-23-07724-f001:**
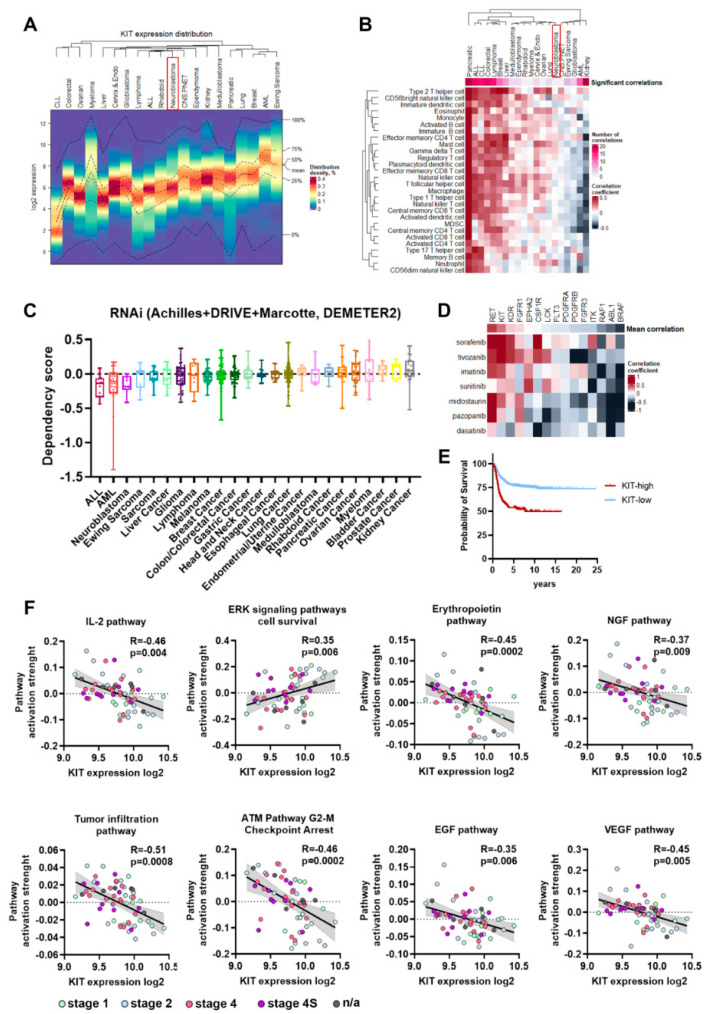
(**A**) Heatmap for KIT expression distribution in 19,145 samples of 20 cancer types. AML-acute myeloid leukemia, ALL-acute lymphoblastic leukemia, CLL-chronic lymphoblastic leukemia. NB is marked by red box. (**B**) Pearson correlations for *KIT* expression and pan-cancer immune infiltration signatures. Number of significant correlations, with *p* < 0.05 after FDR correction and |R| > 0.2 is reflected by upper color scale. Lower color scale reflects Pearson correlation coefficients. NB is marked by red box. (**C**) Dependency scores from the DepMap RNAi screens. Negative scores indicate reduction in cell proliferation after gene knockdown. Cancer cell lines group by their type, each dot represent results for a cell line. (**D**) Pearson correlation between drug AUC values from PRISM database and drug targets dependency scores from DepMap RNAi and CRISPR screens. Mean correlation coefficients for a drug targeted were averaged for all multikinase inhibitors. Drug targets arranged by their mean correlation coefficients and drugs are arranged by Ward.D2 clustering. (**E**) Kaplan–Meier survival analysis for NB patients with relatively high and low KIT expression in tumors. (**F**) Pearson correlation of signaling pathways activation strength (PAS) with KIT expression for NB tumors. PAS values were calculated for 60 NB tumor samples using Oncobox algorithm.

**Figure 2 ijms-23-07724-f002:**
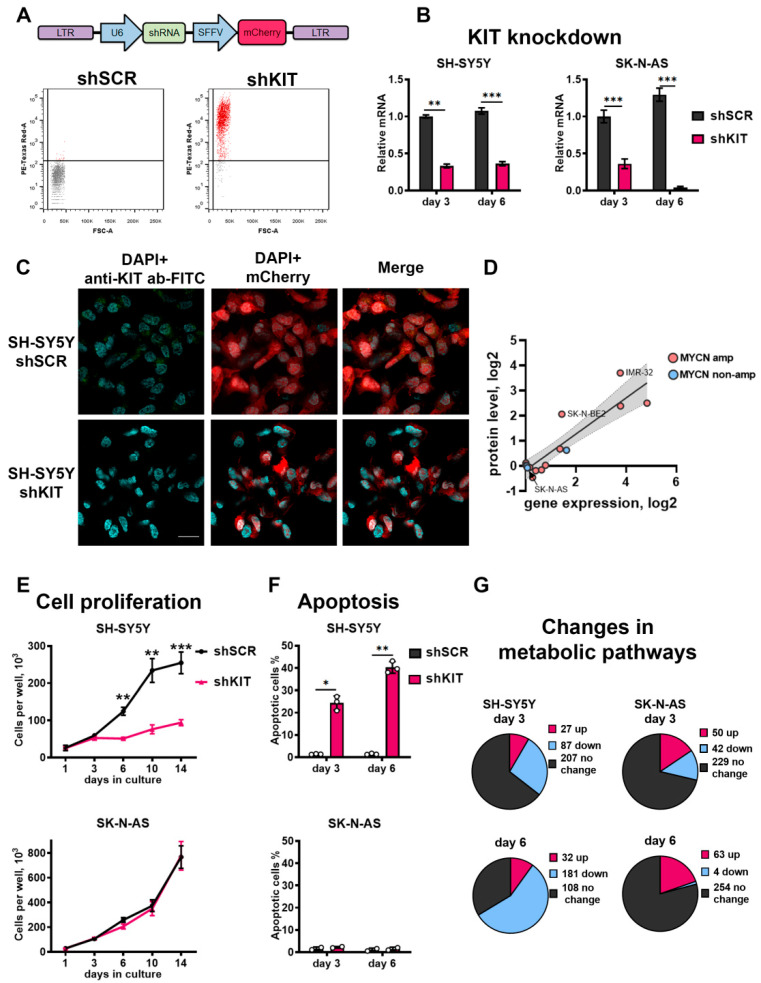
(**A**) Transduction of SH-SY5Y cells with lentiviral vectors encoding control shRNA (shSCR) and shRNA against KIT (shKIT). Transduction efficacy was measured three days after transduction by flow cytometry, based on fluorescence intensity of mCherry protein (PE-Texas Red-A). (**B**) Relative mRNA KIT levels measured by real-time PCR three and six days after lentiviral shRNA transduction. *KIT* gene expression for each sample was normalized to *GAPDH* expression. (**C**) Immunocytochemistry analysis for KIT protein in SH-SY5Y cells 3 days after transduction with shSCR or shKIT lentiviral vectors. Nuclei were stained with DAPI. Larger images are available on Appendix A. (**D**) Correlation between KIT mRNA and protein levels for NB cells according to CCLE gene expression and protein array data. (**E**) Number of viable cells measured for 14 days after transduction with shRNAs. (**F**) Percentage of apoptotic cells measured three and six days after transduction by flow cytometry using annexin-V and SYTOX-blue staining. (**G**) Number of upregulated and downregulated metabolic pathways after transduction of SH-SY5Y and SK-N-AS cells. * *p* < 0.05, ** *p* < 0.01, *** *p* < 0.001 as determined by two-tailed *t*-test for PCR experiments, and by Mann–Whitney test for other experiments.

**Figure 3 ijms-23-07724-f003:**
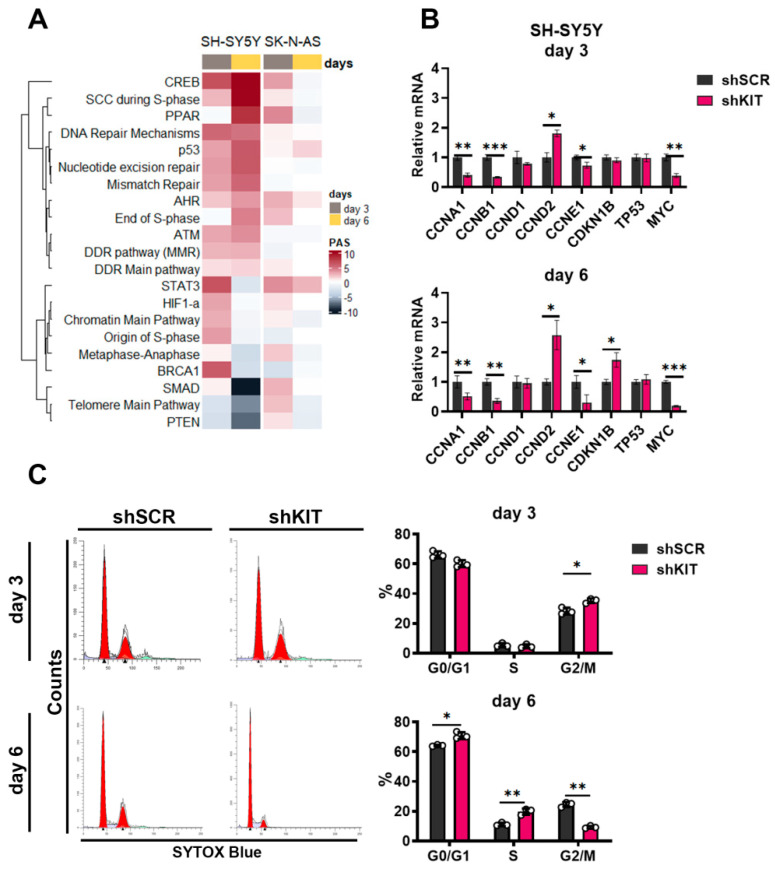
(**A**) Pathway activation strength for pathways related to cell cycle progression and DNA stability. Pathways were clustered using Ward’s clustering method. (**B**) Relative mRNA levels for cell cycle regulators measured by real-time PCR three and six days after lentiviral shRNA transduction of SH-SY5Y cells. Each gene expression for each sample was normalized to *GAPDH* expression. (**C**) Cell cycle phase distribution for SH-SY5Y three and six days after transduction measured by flow cytometry using SYTOX-blue DNA staining. Percentages calculated with ModFit LT 5.0 are provided for G0/G1, S, and G2-M phases. * *p* < 0.05, ** *p* < 0.01, *** *p* < 0.001 as determined by two-tailed *t*-test for PCR experiments, and by Mann–Whitney test for other experiments.

**Figure 4 ijms-23-07724-f004:**
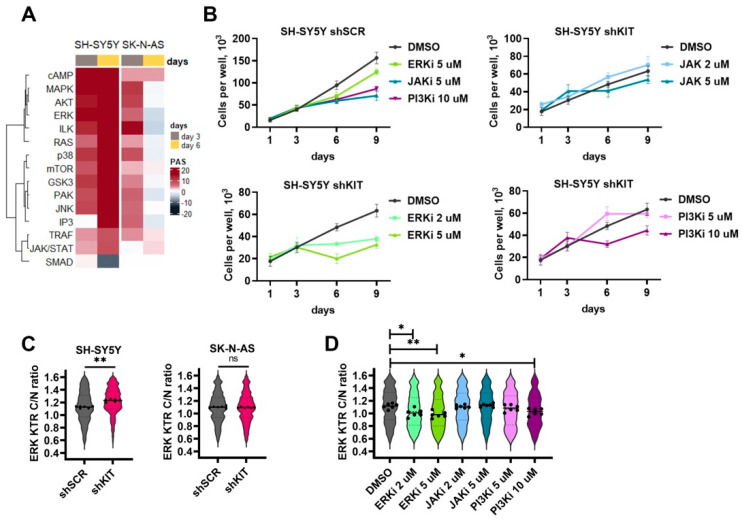
(**A**) Pathway activation strength for pathways related to intracellular kinase activity. Pathways were clustered using Ward’s clustering method. (**B**) Number of viable cells transduced with shSCR or shKIT in the presence of ERK1/2 inhibitor FR180204 (ERKi), JAK2 inhibitior AG490 (JAKi), or PI3K inhibitor wortmannin (PI3Ki). Drugs were added on day 3, then growth medium was changed on day 6, and drugs were added in the same concentrations. (**C**) ERK activity distribution in SH-SY5Y and SK-N-AS cells three days after transduction with shSCR or shKIT vectors. ERK activity was determined by calculation of cytoplasm to nuclei (C/N) ration for ERK-KTR reporter. Dots represent median C/N ratios for each of six biological repeats. Each repeat contains data for 400-700 cells. Violin plots show C/N ratio distribution for all analyzed cells. (**D**) ERK activity distribution for SH-SY5Y cells treated with kinase inhibitors for 24 h. * *p* < 0.05, ** *p* < 0.01 as determined by Mann–Whitney test.

**Figure 5 ijms-23-07724-f005:**
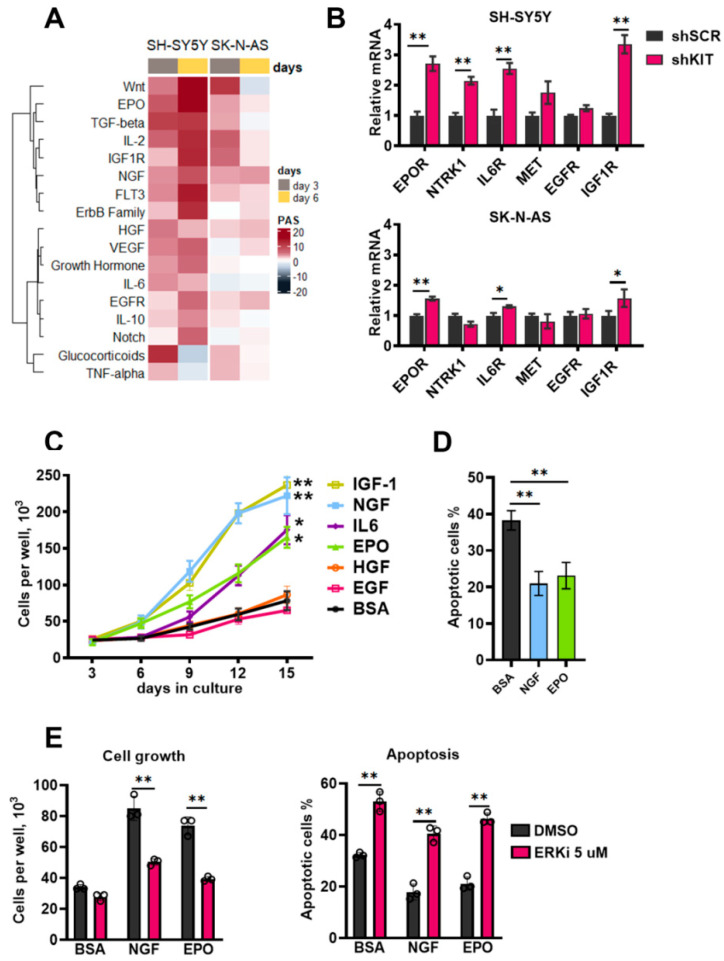
(**A**) Pathway activation strength for pathways related to growth factor signaling. Pathways were clustered using Ward’s clustering method. (**B**) Relative mRNA levels for growth factor receptor genes measured by real-time PCR six days after lentiviral shRNA transduction of SH-SY5Y and SK-N-AS cells. Each gene expression for each sample was normalized to *GAPDH* expression. (**C**) Number of viable cells measured for 15 days after SH-SY5Y transduction with *KIT*-specific shRNAs in the presence of recombinant growth factors. Recombinant growth factors (100 ng/mL) were added on day 3 after lentiviral transduction, then on day 6, growth medium was changed and growth factors were added in the same concentrations, and this process was repeated on days 9 and 12. Bovine serum albumin (BSA) was used as a control mock treatment, as it was used for protein stabilization. (**D**) Percentage of apoptotic cells measured six days after transduction. EPO and NGF (100 ng/mL) were added on the third day after transduction. Apoptosis was measured by flow cytometry using annexin-V and SYTOX-blue staining. (**E**) Number of viable cells and percentage of apoptotic cells six days after SH-SY5Y transduction with shKIT lenviral vector in the presence of EPO and NGF and ERK1/2 inhibitor FR180204 (ERKi). EPO and NGF (100 ng/mL) and FR180204 (5 μM) were added on the third day after transduction. * *p* < 0.05, ** *p* < 0.01 as determined by two-tailed *t*-test for PCR experiments, and by Mann–Whitney test for other experiments.

## Data Availability

All data not included in Appendix A are available on request. Codes are available at https://github.com/CancerCellBiology/NB-shKIT-paper, accessed on 5 July 2022.

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
