# Peer review of "Subtype of Neuroblastoma Cells with High KIT Expression Are Dependent on KIT and Its Knockdown Induces Compensatory Activation of Pro-Survival Signaling"

_ijms, 2022, doi:10.3390/ijms23147724_

Round 1

Reviewer 1 Report

Dear authors, my question cannot serve as an obstacle to delaying the article. I did not find in your works ideas about the subtle mechanisms of switching the cell to the oncoprocess mode. For me personally, it is interesting what chemical processes are triggered. Previously authors showed that RTK inhibitors, such as imatinib, cause increased expression of several growth factor receptors, including EPOR and TRKA, and elevate ERK1/2 activity in NB cells. The authors of the article have ideas about the subtle mechanism of action RTK inhibitors on KIT?

Author Response

We thank the Reviewer for dedicating their time to review our manuscript. Indeed this is an interesting question on how switch in RTK expression happens in cancer cells. Although we do not have strong experimental evidence on the exact mechanisms, our best guess that such switch involves some strong transcriptional factors such as MYC. We included the following speculations in Discussion section:

“Previously we showed that imatinib and some other RTK inhibitors induce changes in growth factor receptors expression similar to KIT knockdown [21]. Here we also found strong decrease in MYC expression after KIT knockdown (Figure 3B). MYC is a potent regulator for many receptors gene transcription. A study by Duncan J. et al. shows that inhibition of RTK signaling including ERK1/2 leads to MYC degradation, which triggers kinome reprogramming and increase of RTKs expression [52]. This then leads to reactivation of ERK1/2 activity and cell adaptation to initial stress. Also MYC/MAX complex may act as a component of a negative feedback control of ERK1/2 activity. MYC is activated by ERK1/2 phosphorylation [53] and MYC/MAX complex activates expression of dual-specificity phosphatases DUSP2 and DUSP7 [54], which negatively regulate ERK1/2 phosporylation. Also many growth factor receptors have common regulatory proteins, which connect them with MYC and MAPKs as shown by GeneMania analysis (Figure S6). These adaptor proteins might also be involved in RTK reprogramming. However, these speculations on mechanisms of kinome reprogramming in NB cells are not experimentally validated and need further experimental confirmation.”

Reviewer 2 Report

Understanding the clinical and biological significance of the c-kit proto-oncogene in a variety of tumors including neuroblastomas(NBs) is a challenging task. Interestingly there has been much progress in the availability of innovative tools including pharmacological agents to alter c-kit expression in a variety of cellular models. In this context, the article entitled « Neuroblastoma cells are dependent on KIT expression and its knockdown induces compensatory activation of pro-survival signaling » is potentially attractive.  This article is an extension of previous studies by the authors, and relies on 2 parts, an observational part and a mechanistical part.  

  • The title is potentially misleading as some NBs including cell lines express c-kit at a low level. 
  • Regarding the observational part of this article, it is unfocused and confusing. In addition it exploits several datasets of dissimilar nature. Therefore the informational content of this part is low.  In fact there is no attempt to establish clinico-pathological correlations in connection with the level of c-kit expression specifically in NBs. Such correlations ( age of patients, NBs pathological classification, MYCN amplification, staging) are important to shed light on the potential role of c-kit expression (see for example Krams M et al. Oncogene 2004) . In addition evaluation of c-kit expression only at the mRNA level may be misleading. 
  • The mechanistic part of this article  aims at reducing the expression of c-kit at the mRNA level solely based on a sh-RNA strategy. It relies only on 2 cell lines whose phenotypic features are not described. Other strategies are available with a translational relevance, that take advantage  of monoclonal antibodies (mAbs) that bind cell-surface CD117 (c-Kit) leading to a functional blockade of c-kit receptor. The authors should be encouraged to explore this strategy. Interestingly the authors show that « NGF, IGF-1 and EPO were able to increase cell proliferation in KIT depleted cells in ERK1/2-dependent manner ». Such a finding should be extended to a bunch of cell lines. 

Author Response

We thank the Reviewer for dedicating their time to review our manuscript and providing thoughtful comments. We agree that title might be misleading and we adjusted it, so it reflects that not all NB cells are KIT dependent: “Subtype of neuroblastoma cells with high KIT expression are dependent on KIT and its knockdown induces compensatory activation of pro-survival signaling.”

In this study we aimed at characterization of KIT expression in NB cells on different levels and thus used datasets of different nature, including six datasets contacting data for NB tumors, and three for NB cell lines. We believe that showing similar aspects of KIT functions in different independent datasets increases reliability and makes our data more reproducible. We agree that using different datasets might be confusing and we included more detailed description to the Results part and dataset description to Table S1.

The clinico-pathological aspects correlations for KIT in NB tumors has been described in several previous publications and we mostly focused on investigating it’s association with cellular processes. But we agree with the Reviewer that description of KIT clinico-pathological associations should be added to the manuscript. We included description of these analysis and verified them using larger NB dataset. The following sentences were added to the Results:

“As was previously shown KIT expression is higher in MYCN-amplified and INSS stage 4 tumors, and in patients older than 18 months [6, 8, 9, 21]. These patterns of KIT expression were confirmed by analysis of integrated Cangelosi dataset (Figure S1).”

Also as the Reviewer pointed out there were several studies regarding KIT clinico-pathological correlations, including measuring KIT at protein levels. This is an important issue raised by the Reviewer, as there were contradictions in previous studies on KIT correlations with tumor stages and outcome. We included the description of previous studies in the Discussion section:

“Early histological indicated that higher KIT content in NB tumors might be associated with less aggressive tumors and more favorable prognosis [42-44]. However later studies showed strong association of KIT with cancer stem cells, stage 4 and MYCN-amplified tumors [6, 8, 9, 14, 16, 45, 46], and our data strongly supports these findings.”

As the Reviewer pointed out measuring KIT at mRNA level only might be misleading. This is a valid point and we agree that sometimes changes at mRNA level do not translate to protein level. Previously we showed that NB cells express KIT and mRNA and protein levels and they correspond well. Here we added immunocytochemistry analysis to show that KIT knockdown also affects KIT protein levels. Also previous studies show that KIT is expressed in NB tumors at protein level and our analysis of CCLE data showed strong correlation between KIT mRNA and protein expression in NB cells. The following was added to the Results section:

“KIT knockdown in SH-SY5Y cells was further confirmed by immunocytochemistry analysis (Figure 2C, S3). KIT is expressed at protein level in NB tumors [8], and we previously showed that KIT mRNA levels had good association with KIT protein levels in NB cells [7]. These data are supported by strong correlation between KIT mRNA and protein levels in NB cells according to CCLE data (Figure 2D).”

The cell lines used in this study are commonly used NB cells and are well characterized. We added description of cell lines to the Methods section.

We agree that adding data for additional growth factor action on additional cell lines sensitive to KIT knockdown would improve reliability of our data. We added the data that we had for two additional cell lines:

“To verify these findings we selected two additional NB cell lines with relatively high KIT expression: SK-N-BE(2) and IMR-32 (Figure 2D). Both cell lines were sensitive to KIT knockdown and EPO rescued both SK-N-BE and IMR-32 (Figure S5). EPOR, NTRK1, IL6R, and IGF1R expression was upregulated in SK-N-BE(2) cells (Figure S5). Due to high toxicity of KIT knockdown for IMR-32 cells we were not able to measure gene expression changes in these cells.”

Performing additional experiments with KIT-targeting antibodies would improve the understanding whether KIT is a good therapeutic target or not. However in this study we mainly focused on KIT functions in various signaling pathways and testing if the effects after KIT knockdown would mimic the action of multikinase inhibitors, which we described previously. Unfortunately we do not have an opportunity to perform additional experiments in 10 days given for this revision.

Round 2

Reviewer 2 Report

Generally speaking, the authors took heed of the remarks raised by the reviewer (some typos left, see for example line 77). However I have a concern regarding Fig2 C.  The micrographs cannot be interpreted because of poor definition. In addition, 2 micrographs, upper and lower panels on the right  (merge): seems to be the same picture.

Author Response

Dear Reviewer,

We are grateful that you noticed the mistake in Figure 2. Indeed micrographs in Figure 2C were in the wrong order, as we inserted the wrong figure version in the manuscript by mistake. We uploaded the right Figure 2 version with micrographs in 2C in the right order and with a scale bar. Merge and mCherry images may look similar due to moderate FITC intensity compared to mCherry. Also, Figure 2D panel was missing and in the new figure version we added this panel. Micrographs were inserted using the maximum quality, no compression was used. To provide larger images we included Figure S3, which also contains cells stained with isotype control antibodies. We also corrected typos throughout the manuscript. 

We are sorry for the confusion caused by the wrong figure and again we are grateful for Reviewer's careful examination of our manuscript. We attach the Figure S3 with larger micrographs, which is also present in the supplementary file.

Round 3

Reviewer 2 Report

Fig.2 C is better now, after corrective action. Despite commendable efforts and improvement, it is still not very convincing (pixelization), probably due to some technical limitations. If one looks at the Fig2 through the eyes of faith, it could be OK! Fortunately the authors have more convincing arguments to support their results and interpretations.